# Enabling High Activity Catalyst Co_3_O_4_@CeO_2_ for Propane Catalytic Oxidation via Inverse Loading

**DOI:** 10.3390/molecules28155930

**Published:** 2023-08-07

**Authors:** Xuan Wang, Wei Liang, Changqing Lin, Tie Zhang, Jing Zhang, Nan Sheng, Zhaoning Song, Jie Jiang, Bing Sun, Wei Xu

**Affiliations:** 1SINOPEC Research Institute of Safety Engineering Co., Ltd., 339th Songling Road, Qingdao 266071, China; wangxuan.qday@sinopec.com (X.W.); liangw.qday@sinopec.com (W.L.); zhangt.qday@sinopec.com (T.Z.); zhangjing.qday@sinopec.com (J.Z.); jiangj.qday@sinopec.com (J.J.); sunb.qday@sinopec.com (B.S.); 2School of Physical Science and Technology, Guangxi University, Nanning 530004, China; 2007401043@st.gxu.edu.cn; 3Department of Chemical Engineering, Tsinghua University, Beijing 100084, China

**Keywords:** propane catalytic oxidation, inverse loading, oxygen vacancy

## Abstract

Propane catalytic oxidation is an important industrial chemical process. However, poor activity is frequently observed for stable C–H bonds, especially for non-noble catalysts in low temperature. Herein, we reported a controlled synthesis of catalyst Co_3_O_4_@CeO_2_–IE via inverse loading and proposed a strategy of oxygen vacancy for its high catalytic oxidation activity, achieving better performance than traditional supported catalyst Co_3_O_4_/CeO_2_–IM, i.e., the T_50_ (temperature at 50% propane conversion) of 217 °C vs. 235 °C and T_90_ (temperature at 90% propane conversion) of 268 °C vs. 348 °C at the propane space velocity of 60,000 mL g^−1^ h^−1^. Further investigations indicate that there are more enriched oxygen vacancies in Co_3_O_4_@CeO_2_–IE due to the unique preparation method. This work provides an element doping strategy to effectively boost the propane catalytic oxidation performance as well as a bright outlook for efficient environmental catalysts.

## 1. Introduction

Environmental pollution, especially atmospheric environment pollution, is becoming an increasingly serious problem. Volatile organic compounds (VOCs) emission gained widespread attention in very recent years [1,2,3]. The term VOCs means any compound of carbon, excluding carbon monoxide, carbon dioxide, carbonic acid, metallic carbides or carbonates, and ammonium carbonate, which has become a main pollutant in the whole world. VOCs come from manifold sources, such as the petrochemical industry, construction materials, the printing industry and the electronics industry. In addition to the direct health effects, such as mutagenesis, teratogenesis and carcinogenesis, VOCs form 2.5 micrometer particulate matter (PM2.5) and damage the ozone layer through a series of photochemical reactions, thus seriously affecting the air quality around our living environment [4,5,6].

Propane, representing typical VOCs, is widely released from the energy activities involving liquefied natural gas (LNG), compressed natural gas (CNG) and liquefied petroleum gas (LPG), as well as other important industrial processes [7,8]. The effective treatment process of propane is catalytic oxidation for it provides high performance at a relatively low reaction temperature. However, there is still a problem in the catalytic oxidation of propane at low temperatures due to the stable C–H bonds in propane. Generally, precious metal (e.g., Au, Pd, Pt, Ru) catalysts show high activity in the propane catalytic oxidation [9,10,11,12]. Although noble metal catalysts, such as Pd and Pt, are supposed to be highly active and stable, their high expense, sintering rates, volatility, and the possibility of being poisoned by water or sulfur compounds limit their wide practical application. With gradually increasing awareness of environmental protection and strict emission control regulations, highly efficient cleaning technologies of dilute propane are in great demand. More attention has been paid to transition metal oxides (such as manganese, cobalt, and copper oxides) in recent years [13,14,15,16,17,18]. Co_3_O_4_ and its composite oxides emerge to be competitive alternative catalysts for noble metal catalysts due to their advantages, such as varied Co valence (Co^2+^ and Co^3+^), unfilled *d* orbitals, redox property, affluent active oxygen, rich resources and low prices. And it is one of the most effective active species in the cleavage of C–C and C–H bonds [19,20,21]. In the past decades, numerous efforts have been taken to improve the catalytic activity of Co_3_O_4_ based materials, for example, by controlling the size, dispersity and shape of the nanoparticles, the exposed crystal faces and the defect concentration of the surface [22,23,24]. New active sites can be constructed by surface engineering on the defects of Co_3_O_4_ based materials to regulate catalytic activity.

Oxygen vacancies (O_V_), one type of common defects, were first proposed in the 1960s [25]. O_V_ can capture oxygen from the surrounding atmosphere and convert gaseous oxygen molecules into more reactive oxygen species. The mobility and activity of lattice oxygen species can be improved via the transmission effect of bulk vacancies. Later, researchers discovered that O_V_ can be used as reaction sites to change the structure of the material and the electronic and chemical properties of the surface [26,27,28]. CeO_2_ acts as an attractive support for the catalyst to generate O_V_ when Ce^4+^ is reduced to Ce^3+^ that corresponds to Ce_2_O_3_ and CeO_2_. As the most abundant rare earth element, its content in the crust is 0.0046%. It is an acidic substance with plentiful Lewis acid sites and a few Brønsted acid sites. Acidic sites can improve the mobility of reactive oxygen species, enhance the adsorption of propane, promote the breaking of C–H bonds and thus contribute to catalytic oxidation of propane [29,30,31,32]. Moreover, O_V_ in the support can improve the dispersion and stability of catalysts by anchoring the active metal. However, as far as we know, there are few reports about the simple method for O_V_ of the Co_3_O_4_−based catalyst for the propane catalytic oxidation. Thus, it is very necessary to develop a concise route to construct the O_V_ of the catalyst for high catalytic activity and low temperatures of propane catalytic oxidation.

In this study, we designed an inverse loading method, called ion−exchange method, and accordingly synthesized a catalyst Co_3_O_4_@CeO_2_–IE (Figure 1) possessing unique structure different from the traditional supported catalyst Co_3_O_4_/CeO_2_–IM synthesized by impregnation method. For traditional impregnation (IM) methods, an inorganic salt precursor, Co(NO)_3_∙6H_2_O is used as the starting precursor to deposit the active species onto the outer surface of CeO_2_. These loading methods normally render limited contact area/density and relatively weak interfacial interaction between the active species and the support, resulting in easy aggregation of active species, simultaneously sharp reduction in the interfacial area/density during the pretreatment or reaction process and further impact on the catalytic performance (Figure 1a). In contrast, for the IE reaction, the Co(OH)_2_ nanosheet precursor was added into the solution of support CeO_2_ precursor Ce^3+^ solution and the ion–exchange (IE) reaction took place driven by the difference of solubility product (K_sp_), as shown in Figure 1b. Co^2+^ was replaced with Ce^3+^ to form Co(OH)_2_@Ce(OH)_4_ mixed metal hydroxide (MMH) structure. The Co_3_O_4_@CeO_2_–IE catalyst was obtained by the subsequent calcination of Co(OH)_2_@Ce(OH)_4_ MMH. Co_3_O_4_@CeO_2_–IE exhibits a T_50_ of 217 °C and T_90_ of 268 °C at propane space velocity of 60,000 mL⋅g^−1^⋅h^−1^, comparing to T_50_ of 235 °C and T_90_ of 348 °C for the Co_3_O_4_/CeO_2_–IM. The investigations indicate that the enriched oxygen vacancies resulting from unique synthetic method, i.e., ion–exchange method, increase the propane catalytic oxidation efficiency.

## 2. Results

### 2.1. Structure and Morphology of the Obtained Samples

Two sets of composite samples with different loading directions, i.e., the Co_3_O_4_@CeO_2_–IE and Co_3_O_4_/CeO_2_–IM, and similar chemical compositions were synthesized (Appendix A). The morphology and hierarchical structure of the composites were characterized by a high–resolution transmission electron microscope (HRTEM) and energy dispersive spectrometer (EDS) elemental mapping analyses. The transmission electron microscope (TEM) images of Co(OH)_2_ show ultrathin nanosheets (about 10 nm) with clear lattice spacing of 0.26 nm (Appendix A). And the typical HRTEM images (Figure 2a) of Co_3_O_4_@CeO_2_–IE shows the CeO_2_ and Co_3_O_4_ particles are uniformly distributed. As shown in Figure 2b, the typical HRTEM images clearly show the Co_3_O_4_ (311), (220) and CeO_2_ (111) with interlayer spacing of 0.243 nm, 0.288 nm and 0.310 nm, respectively. EDS elemental mapping further proves the homogeneous distribution, implying a sufficient contact between Ce and Co of the Co_3_O_4_@CeO_2_–IE catalyst. (Figure 2c). In comparison, Co_3_O_4_/CeO_2_–IM shows a heterogeneous distribution between Co and Ce, and the CeO_2_ is reunited, meaning that there is a weak interaction between Co_3_O_4_ and CeO_2_ (Figure 2d–f). These results suggest that Co_3_O_4_@CeO_2_–IE mainly exposes the CeO_2_ inserts into Co_3_O_4_ surface, while Co_3_O_4_/CeO_2_–IM mainly stands on the surface of Co_3_O_4_. And the particle size of Co_3_O_4_@CeO_2_–IE (10.8 nm) and Co_3_O_4_/CeO_2_–IM (41.8 nm) was calculated by Scherrer equation, indicating that the smaller particles could be fabricated by ion–exchange methods.

The phase composition and crystallographic structure of Co_3_O_4_@CeO_2_–IE samples and Co_3_O_4_/CeO_2_–IM were examined by powder X-ray diffraction (PXRD). Figure 3a shows PXRD patterns Co_3_O_4_@CeO_2_–IE and Co_3_O_4_/CeO_2_–IM, which can be indexed to the composite phases of Co_3_O_4_ (JCPDS:42–1467) and CeO_2_ (JCPDS:34–0394), further confirming the similarity in the effect of IE and IM methods on the preparation of the catalyst. However, there is a discrepancy in the intensity of the diffraction peaks. Co_3_O_4_/CeO_2_–IM has higher intensity of the diffraction peaks thanCo_3_O_4_@CeO_2_–IE, suggesting that the former has better crystallinity than the latter and the uniform distribution affects the crystallinity of Co_3_O_4_ and CeO_2_. Notably, the characteristic peaks of Co_3_O_4_@CeO_2_–IE shift to lower 2θ values due to the diffusion of Ce^4+^ and Co^2+^, Co^3+^ into the opposite crystal lattice at the interface of Co_3_O_4_ and CeO_2_ while smaller Co^2+^ (74.5 pm) and Co^3+^ (61 pm) than Ce^4+^ (97 pm) [33], thus forming a mixed phase. Co_3_O_4_/CeO_2_–IM has almost the same peak positions as Co_3_O_4_ and CeO_2_, indicating that interactions between the Co_3_O_4_ and CeO_2_ seldom occur at the interface of Co_3_O_4_/CeO_2_–IM, consistent with the results of HRTEM analysis. The analysis results of nitrogen adsorption isotherms of the catalysts demonstrate that Co_3_O_4_@CeO_2_–IE (94.7 m^2^ g^−1^) has a much higher BET surface area than Ni–CeO_2_–IM (24.3 m^2^ g^−1^), meaning that the IE method can expose more active sites for the following catalytic applications (Figure 3b).

### 2.2. Catalytic Oxidation Performance

The propane catalytic oxidation test was performed in the fixed bed reactor at WHSV of 60,000 mL⋅g^−1^⋅h^−1^. The oxidative reactivity was evaluated by T_50_ and T_90_. Figure 4a shows that the Co_3_O_4_@CeO_2_−IE has much higher activity than Co_3_O_4_/CeO_2_–IM. Co_3_O_4_@CeO_2_−IE as it participates in the propane catalytic oxidation reaction, with T_50_ of 217 °C and T_90_ of 268 °C, significantly superior to T_50_ of 235 °C and T_90_ of 348 °C for Co_3_O_4_/CeO_2_–IM catalyst. The activation energies (E_a_) were evaluated according to the Arrhenius plots in Figure 4b, demonstrating a much lower Ea value of 63.7 kJ⋅mol^−1^ for Co_3_O_4_@CeO_2_−IE than Co_3_O_4_/CeO_2_−IM (89.5 kJ⋅mol^−1^). And Co_3_O_4_@CeO_2_–IE shows a superior activity toward the total oxidation of propane to Co_3_O_4_/CeO_2_–IM, and its reaction rate at 235 °C is 9.80 × 10^−7^ mol g^−1^ s^−1^, is almost 1.5 times that of Co_3_O_4_/CeO_2_–IM (6.91 × 10^−7^ mol g^−1^ s^−1^). These findings confirm that Co_3_O_4_@CeO_2_−IE exhibits higher catalytic activity (Appendix A).

### 2.3. Surface Chemistry Analysis of the Catalysts

To gain deep insight into the origin of differences in catalytic behaviors between Co_3_O_4_@CeO_2_–IE and Co_3_O_4_/CeO_2_–IM, X-ray photoelectron spectroscopy (XPS) analysis was used to access the electronic state of the catalyst. The Ce 3d XPS spectra for the two catalysts were performed. And the spectra were deconvoluted into ten sub−peaks. The peaks denoted as U and V are individually attributed to Ce 3d_5/2_ and Ce3d_3/2_, respectively. Among those ten peaks, U_0_, U^I^, V_0_ and V^I^ are assigned to Ce^3+^ (red line) and the rest of them are related to Ce^4+^ (blue line) [34]. As depicted in Figure 5a, the XPS results reveal that the Ce^3+^ fraction (49.6%) on the surface of Co_3_O_4_@CeO_2_–IE is nearly twice more than that of Co_3_O_4_/CeO_2_–IM (28.6%). It is proposed that there are more oxygen defects on the surface of Co_3_O_4_@CeO_2_−IE (Ce^4+^ + O_L_ → Ce^3+^ + O_V_) [35,36]. The relative oxygen vacancy (O_v_) proportion on the surface of Co_3_O_4_@CeO_2_–IE (40.7%) is much higher than that of Co_3_O_4_/CeO_2_–IM (25.0%). The peaks at 780.5 eV and 795.8 eV are attributed to the Co 2p_1/2_ and Co 2p_3/2_ core line of Co^2+^, respectively, whereas those at 779.2 eV and 794.2 eV are related to Co^3+^ [37]. The ratio of Co^2+^/Co^3+^ (2.64) in Co_3_O_4_@CeO_2_–IE is much higher than that (1.96) in Co_3_O_4_/CeO_2_−IM due to the fact that the oxygen defects of the catalyst can promote the reduction in metal ions (Figure 5c and Appendix A).

The structures of oxygen vacancies were further investigated by electron paramagnetic resonance (EPR). Figure 6a,b show that the two catalysts have the g value of 2.004, which can be attributed to the unpaired electrons trapped in the O_V_ in the Co_3_O_4_/CeO_2_ materials. Co_3_O_4_@CeO_2_−IE has higher intensity of the peaks than Co_3_O_4_/CeO_2_–IM. The intensity of the peaks, which is associated with the number/density of the O_v_, implies that Co_3_O_4_@CeO_2_−IE have more O_v_.

The surface defects were further examined by the Raman spectroscopy (Figure 6c). Co_3_O_4_/CeO_2_−IM exhibits the signals of 191 (F_2g_^1^), 474 (E_g_), 516 (F_2g_^2^), 615 (F_2g_^3^) and 680 cm^−1^ (A_1g_), which correspond to pure Co_3_O_4_ [38], indicating that Ce seldom interacts with Co_3_O_4_ in Co_3_O_4_@CeO_2_−IE (in accordance with the results of HRTEM and XRD analysis). All of the bands should be mainly assigned to the vibration mode of Co_3_O_4_. The sharp band at 461 cm^−1^ (F_2g_ band) of CeO_2_ can be assigned to the vibration mode of CeO_2_ fluorspar structure (Appendix A) [39,40]. The F_2g_ band of Co_3_O_4_@CeO_2_−IE shows a red−shift of 16 cm^−1^ to 445 cm^−1^ with a sharp peak, which can be attributed to the Co–O–Ce bond induced by residual stress or lattice distortion in CeO_2_ structure, further suggesting a strong interaction between Co_3_O_4_ and CeO_2_ in Co_3_O_4_@CeO_2_−IE [41].

O_2_−TPD profiles were determined to recognize the oxygen species desorbed from the surface as a function of temperature. Figure 6d shows more obvious adsorption (225 °C–600 °C) and (725 °C–950 °C) of oxygen species in Co_3_O_4_@CeO_2_−IE than in Co_3_O_4_@CeO_2_−IM, indicating more O_V_. The quantitative analysis shows that Co_3_O_4_@CeO_2_−IE has nearly three times higher desorptive capacity of O (0.0838 mmol g^−1^) than Co_3_O_4_/CeO_2_−IM (0.0284 mmol g^−1^). The defects O_V_ caused by Ce doping can strengthen the adsorption, activation ability of oxygen molecules and surface oxygen species migration ability, resulting in the generation of abundant active oxygen species at low temperatures.

The above analysis results indicate that the Ce^3+^ ions are partly exchanged with the Co^2+^ ions in Co(OH)_2_, leading to a special interface between Co(OH)_2_ and Ce(OH)_4_ with the diffusion of Ce^4+^ and Co^2+^ into opposite crystal phases and thus more O_v_ of the interface in Co_3_O_4_@CeO_2_−IE via inverse loading. There existed the lattice distortion due to the radius of Co^2+^ or Co^3+^ being smaller than Ce^4+^ and the charge imbalance for Co^2+^ or Co^3+^ and Ce^4+^ in Co_3_O_4_@CeO_2_−IE, that may induce the more active oxygen species of Co–O–Ce in the interface than in the pure CeO_2_ which induced the formation of more interfacial oxygen vacancies, while the Co_3_O_4_ and CeO_2_ in Co_3_O_4_/CeO_2_−IM exhibited poor interaction.

### 2.4. Catalytic Mechanism and Density−Functional Theory (DFT) Calculations

Marse−van Krevelen (MvK) mechanism is suitable for most non−metal catalysts to oxidize VOCs. MvK mechanism is based on redox reaction. Its essence is that the lattice oxygen in the catalyst oxidizes VOCs. In this reaction process, firstly, when VOCs react with lattice oxygen and generate oxygen vacancies, the metal oxides are reduced consequently. Secondly, oxygen vacancies are filled by oxygen in the air. The adsorption capacity and oxygen transfer capacity are the key contributing factors of the MvK mechanism. It is widely accepted that the dissociative adsorption of C_3_H_8_ on the catalyst surfaces triggers the catalytic oxidation process [42,43]. We calculated DFT in order to further elucidate the adsorption mechanism of Co_3_O_4_@CeO_2_−IE. To simulate the sample, a Co_3_O_4_−(311) surface model was created to expose the most crystal faces in the experiments. The Co_3_O_4_−(311) surface, which contains O_V_ (Co–O_V_) adjacent to Co atoms, was simulated first, and then Co_3_O_4_@CeO_2_−IE was simulated by replacing Co atoms with Ce atoms in Co_3_O_4_. In the simulations, VO is adjacent to both Co and Ce atoms (Co–O_V_–Ce). The reaction is initiated by the adsorption of propane to the surface, followed by the dehydrogenation of propane to produce free radicals, which are finally oxidized to CO_2_ and H_2_O, where the desorption of generated propane radicals from the surface of the catalyst is the rate−limiting step for the entire reaction. Hence, the adsorption energies of propane adsorption are studied. Two substrate structures for adsorption are shown in Figure 7a,b. The calculated adsorption free energies (ΔE_ads_) of propane on the two substrates are 0.17 eV and −0.85 eV, respectively. That indicates that propane is not favorable for adsorption on the Co–O_V_ substrate, but is favorable for adsorption on the Co–O_V_–Ce surface. This is mainly due to the fact that Ce atoms have a greater number of outer electrons than Co atoms, and Ce doping in the system not only induces a large number of O vacancy defects, but also increases the free electrons of the system. The electronic density of Ce in the Co_3_O_4_@CeO_2_−IE system near the Fermi level can be analyzed according to the density of states in Figure 7c,d, conductive to electron exchange with propane. The density of states of the Co–O_V_–Ce structure suggests that the additional electrons provided by the Ce atoms are mainly populated above the Fermi energy level 0.3–0.7 eV above the Fermi energy level. In order to better investigate the adsorption property of propane by the extra electrons from Ce atoms, we calculated the crystal orbital Hamilton population (COHP) between Ce and H after adsorption of propane (Appendix A). We found that the extra electrons provided in the Ce atoms coupled with the H atoms favor the adsorption and dehydrogenation of propane [44]. As shown in the charge density difference in Figure 7e,f, the presence of Ce atoms leads to a greater exchange of electron density between the substrate and propane. When Ce fully invades the surface, Co_3_O_4_@CeO_2_−IE has more Co–O_V_–Ce structures than Co_3_O_4_/CeO_2_−IM [45], and is more conducive to the decomposition of propane.

## 3. Discussion

In summary, we successfully fabricated a new kind of Co_3_O_4_@CeO_2_–IE catalyst by ion−exchange method. The obtained Co_3_O_4_@CeO_2_–IE presents superior catalytic activity in the propane catalytic oxidation. The results of XPS, EPR, Raman spectra and DFT calculations etc., indicate that Co_3_O_4_@CeO_2_–IE catalyst possesses abundant oxygen vacancies on the surface of Co_3_O_4_–CeO_2_, which adsorb the propane. This study not only presents a new kind of non–noble metal catalyst for efficient catalytic oxidation of propane, but also highlights a strategy for the design of the oxygen vacancy for an advanced catalyst.

## 4. Materials and Methods

### 4.1. Catalyst Preparation

#### 4.1.1. Preparation of Co(OH)_2_

All chemicals were purchased from Aladdin, China, and used as received.

Synthesis of Co(OH)_2_ nanosheets. According to our previously reported hybridization route [46], the synthetic process for the Co(OH)_2_ was as follows: Commercial MgCO_3_ was calcined at 750 °C for two hours to obtain MgO. Then, the resultant MgO was put into distilled water with a MgO−to−H_2_O mass ratio of 1:10 under stirring for 24 h. Finally, the white Mg(OH)_2_ product was separated by filtration, washed with deionized water and absolute ethanol three times and dried at room temperature. The resultant Mg(OH)_2_ (0.73 g equivalent to 0.0125 mol) was added to 25 mL of an aqueous solution containing 0.0125 mol of Co(NO_3_)_2_⋅6H_2_O. After stirring vigorously for two hours at room temperature, the green Co(OH)_2_ product was separated by filtration, washed with deionized water and ethyl alcohol three times and dried at room temperature overnight.

#### 4.1.2. Preparation of Co_3_O_4_@CeO_2_–IE Catalyst

The Co_3_O_4_@CeO_2_–IE surrounded catalyst was synthesized by ion−change method. The fresh 1.86 g Co(OH)_2_ was added into the 40 mL solution including of 0.22 g Ce(NO_3_)_3_∙6H_2_O and then stirred under room temperature for 30 min. And thus put the mixture into Teflon−lined stainless steel autoclave, sealed, and maintained at 120 °C for 12 h. When cooled to the room temperature, the yellow–greenish products of Co(OH)_2_/Ce(OH)_4_ were separated by filtration, washed with deionized water and ethanol six times, and dried at 80 °C overnight. The hydroxide products were calcinated at 300 °C for 2 h in muffle furnace denoted as Co_3_O_4_@CeO_2_–IE. The Co loading is 45.3 wt % and the Ce loading is 8.36 wt % which was determined by ICP.

#### 4.1.3. Preparation of Co_3_O_4_/CeO_2_–IM Catalyst

The Co_3_O_4_/CeO_2_−IM catalyst was synthesized by wet impregnation method. CeO_2_ support was prepared by reference method. 5.82 g Co(NO_3_)_2_ ∙6H_2_O was added into deionized 5 mL water solution and then added 0.09 g CeO_2_ into the mixture. The slurry was evaporated under stirring at 110 °C until dried thoroughly. And then followed the calcination denoted as Co_3_O_4_/CeO_2_−IM. The Co loading was 45.4 wt % and the Ce loading was 8.38 wt % which was determined by ICP and the similar content as Co_3_O_4_@CeO_2_–IE.

### 4.2. Characterizations

The morphology was characterized using a Hitachi S−4800 scanning electron microscope (SEM). The chemical composition of the solids was determined by an inductively coupled plasma−atomic core line spectroscopy (ICP–AES) (Thermo Fisher iCAP PRO (OES)). XRD patterns were performed on a Rigaku Miniflex 600 using Ni−filtered Cu Ka radiation (k = 0.15408 nm) at 40 Kv and 40 mA, and the scope of data collection was 2θ = 10–80°. The HRTEM images were obtained using a FEI Talos F200X G2 electron microscope operated at 200 kV. Results of element mapping were obtained on a super–x equipped with an Energy Dispersive Spectrometer (EDS). Brunauer–Emmett–Teller (BET) surface area measurement was conducted on a ASAP 2460 instrument. Before the BET surface area measurement, the samples were dried at 300 °C for 4 h under vacuum. Temperature−programmed desorption of O_2_ (O_2_−TPD) was performed on a Tianjin XQ TP−5080B chemisorption instrument with a thermal conductivity detector (TCD). The sample (100 mg) was first at 300 °C for 1 h to remove moisture under the steam (30 mL/min). After cooling to room temperature, 5 vol% O_2_/N_2_ mixture was switched on with a flow rate of 25 mL·min^−1^ at 50 °C for 1 h, and then cooled down to room temperature in the oxidizing atmosphere. Then continuously purging in a He flow for 30 min, the measurement started from room temperature to 950 °C at a heating rate 10 °C·min^−1^ after the baseline of single was stable. Electron paramagnetic resonance (EPR, Bruker A300, Bremen, Germany) were tested by a FA−200 (JES) electron paramagnetic resonance spectrometer. Raman spectra were carried out on Renishaw inVia Qontor with 532 nm of incident light. X−ray photoelectron spectra (XPS) were recorded on Escalab 250Xi using Al Kα radiation (1486.6 eV, 150 W) with binding energies (BEs) calibrated against the C1s peak of adventitious carbon at 284.8 eV.

### 4.3. Catalyst Evaluation

Propane oxidation test was carried out in a fixed–bed reactor using reactant of 0.5 vol% C_3_H_8_ and 21 vol% O_2_, balanced with N_2_. The Weight–Hourly–Space–Velocity (WHSV) was set at 60,000 mL·g^−1^·h^−1^ with reaction temperature increased from room temperature to 500 °C (5 °C/min). The products were in situ tested by a gas chromatograph (Agilent, GC–7890B, Santa Clara, CA, USA). C_3_H_8_ conversion was acquired from the following formulas:$$C_{3}H_{8}\;\mathrm{conversion}\;(\%)=\frac{\left[C_{3}H_{8}\right]in-\left[C_{3}H_{8}\right]out}{\left[C_{3}H_{8}\right]in}\times 100(\%)$$
where in, [C_3_H_8_]_*in*_ and [C_3_H_8_]_*out*_ represent the inlet and outlet C_3_H_8_ concentrations, separately.

To obtain the information of the apparent activation energy (*E_a_*), the reaction was controlled at the kinetic regime (C_3_H_8_ conversion is under 10%). The equation below was used for the calculation of *E_a_* by acquiring the slope:$$\ln k=\frac{{-E}_{a}}{RT}+C$$
where *k* represents the reaction rate constant, *T* is the absolute temperature and *R* stands for the gas constant.

### 4.4. Computational Details

First−principles calculations were carried out using the projector−augmented wave method implemented in the Quantum ESPRESSO based on density functional theory (DFT) [47,48]. The Perdew–Burke–Ernzerhof (PBE) functional of the generalized gradient approximation (GGA) was adopted for electron exchange and correlation interaction [49]. The van der Waals interactions between layers were corrected using the DFT−D3 functional [50]. The ions relaxation was achieved until the force for per atom was less than 0.02 eV/Å and the total energy converged to 10^−5^ eV. A vacuum spacing of 15 Å was used to prevent interaction between adjacent slabs. The change in free energy for the adsorption (ΔE_ads_) of the target by the catalyst substrate is defined by E_q_: ΔE_ads_ = Etotal – Esubstrate – Etarget, where Etotal is the total energy of the catalyst substrate and propane, Esubstrate and Etarget are the energies of the substrate and free target molecule, respectively.

## Figures and Tables

**Figure 1 molecules-28-05930-f001:**
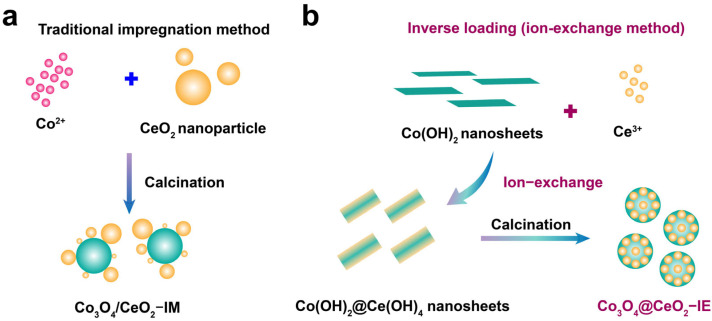
Schematic illustration of the synthesis of (**a**) Co_3_O_4_/CeO_2_–IM prepared by impregnation method and (**b**) Co_3_O_4_@CeO_2_–IE catalyst via inverse loading.

**Figure 2 molecules-28-05930-f002:**
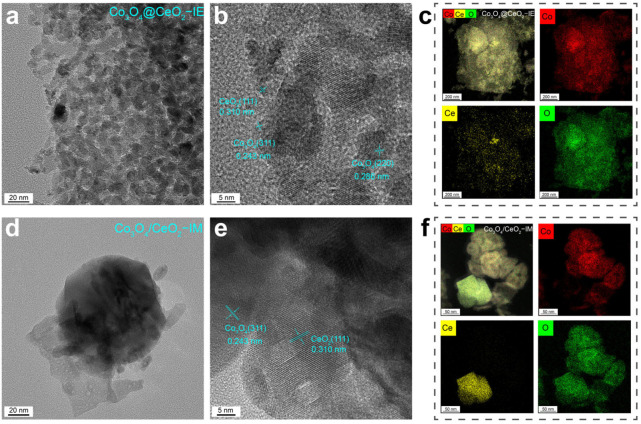
The typical HRTEM images of (**a**,**b**) Co_3_O_4_@CeO_2_–IE catalyst. (**d**,**e**) Co_3_O_4_/CeO_2_–IM catalyst, (**c**) EDS elemental mappings of Co_3_O_4_@CeO_2_–IE and (**f**) Co_3_O_4_/CeO_2_−IM.

**Figure 3 molecules-28-05930-f003:**
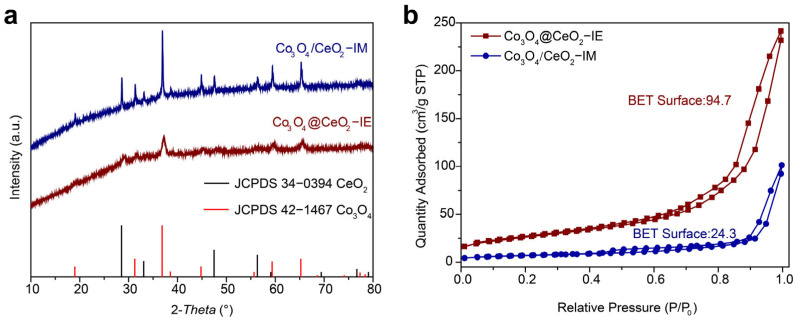
(**a**) Powder XRD patterns and (**b**) N_2_ adsorption/desorption isotherms of Co_3_O_4_@CeO_2_−IE and Co_3_O_4_/CeO_2_−IM.

**Figure 4 molecules-28-05930-f004:**
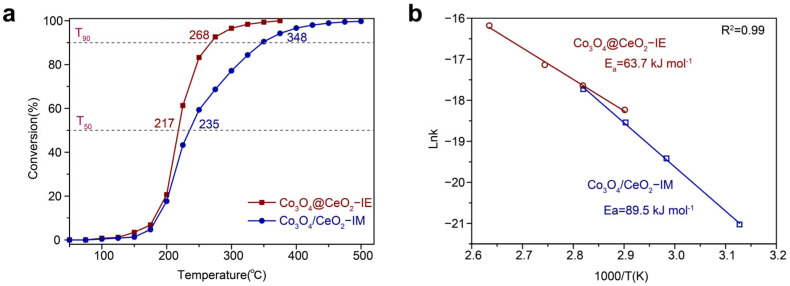
(**a**) Catalytic performance and (**b**) Arrhenius plots of Co_3_O_4_@CeO_2_−IE and Co_3_O_4_/CeO_2_−IM.

**Figure 5 molecules-28-05930-f005:**
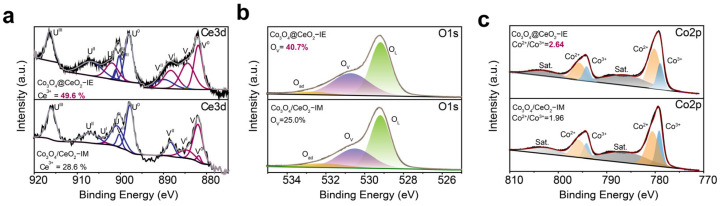
XPS spectra of Ce3d (**a**) (Ce^3+^: V^0^, V^I^, U^0^, U^I^ (red line); Ce^4+^: V, V^II^, V^III^, U, U^II^, U^III^ (blue line)), (**b**) O1s and (**c**) Co 2p for Co_3_O_4_@CeO_2_–IE and Co_3_O_4_/CeO_2_–IM.

**Figure 6 molecules-28-05930-f006:**
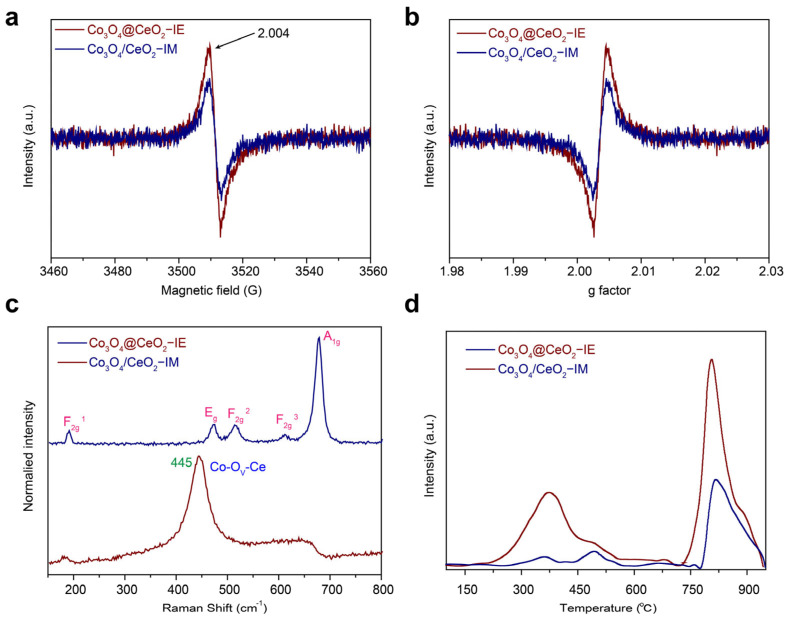
(**a**,**b**) EPR, (**c**) Raman spectra and (**d**) O_2_−TPD profiles of Co_3_O_4_@CeO_2_−IE and Co_3_O_4_/CeO_2_−IM.

**Figure 7 molecules-28-05930-f007:**
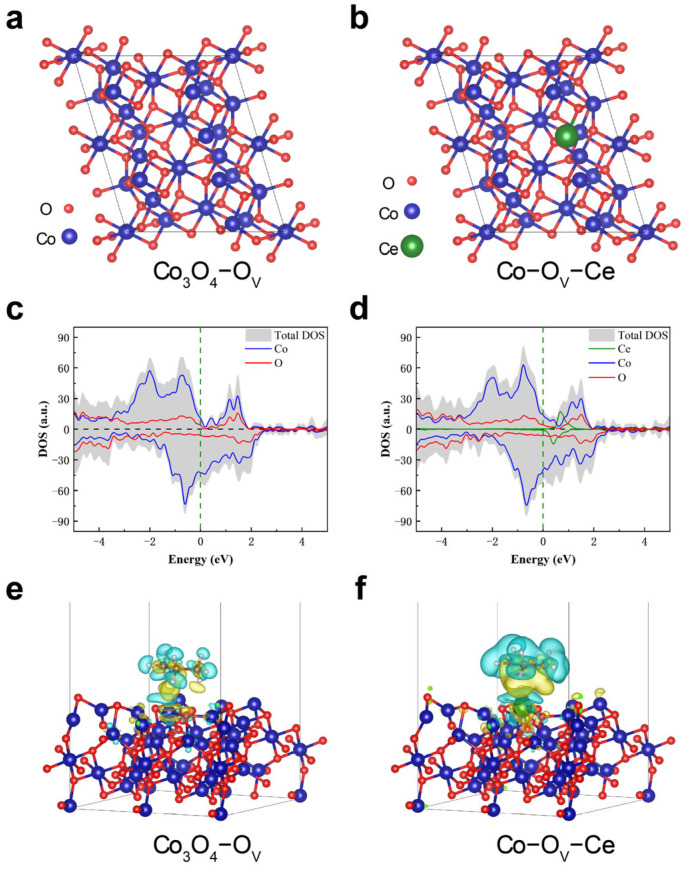
(**a**,**b**) The structures and(**c**,**d**) the electronic density of states (DOS) of the Co_3_O_4_−O_V_ and Co_3_O_4_−O_V_−Ce substrates, respectively. The charge density difference of the substrate after propane adsorption is shown. The green dashed line represents the Fermi energy level (**e**,**f**), where the blue region represents a decrease in charge density and the yellow region represents an increase in charge density. The isosurface level is set at 0.002 e/Å^3^.

## Data Availability

Not applicable.

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
