# Peer review of "Enabling High Activity Catalyst Co3O4@CeO2 for Propane Catalytic Oxidation via Inverse Loading"

_molecules, 2023, doi:10.3390/molecules28155930_

Round 1
Reviewer 1 Report
Comments on Manuscript molecules-2452086
MDPI
Title: Enables Highly Activity Catalyst Co3O4@CeO2 for Propane Cat-2 alytic Oxidation via Inverse Loading
This manuscript present the syntheses of Co3O4@CeO2 proposing a new method IE and compared with the same by the IM. The tests of propane oxidation showed higher activity for the IE catalyst for T50 and T90 . Authors stated that the Ov are determining for this activity and characterized these materials using different techniques.
· In fact, it is a good suggestion and in accordance with the literature the produced sheets favor the higher surface area and metal distribution. The final catalysts showed Ce inserted in Co particles on the IE and Ce anchored on Co particles. However, Co is the active site while Ce oxides promote defects and so increasing the vacancies. In this case, what are the active sites?
· Authors prepared the Co (OH)2 using MgCO3 resulting in Mg(OH)2. Why? Where is Mg in the final product. Authors mention in the abstract the Ni, but no niquel was presented in this manuscript. Moreover, what are the final concentrations of these materials?
· Figure 4a shows the conversions at T50 for both samples IE and IM, but the difference is very small. At T90 there is a significant difference. Important is determining the reaction rate. Please explain?
· The TPO and XPS results were presented qualitatively. Authors didn’t calculate the O release, the Ration of Ce and Co ions from XPS. The O1s showed the same profile but didn’t calculate the adsorbed or lattice oxygen from these results
· The EPR shows at g=2.04 the same peak, but the presence of O would shift this band, indicating defect caused by the interaction of Ce and Co in the structure.
· Figure 4b shows the activation energy, but authors didn’t calculate the rate or k or TOF for determining the E of both IE and IM catalysts. Please explain.
Reviewer 2 Report
The thematic is interesting, however the manuscript need to be improved. In the present manuscript, the authors propose a doping strategy for boosting the propane catalytic oxidation performance. In particular a Co3O4@CeO2-IE material is synthesized, where Ce3/4+ is intended to substitute Co2/3+ sites from a Co(OH)2 nanosheet precursor. From the XRD pattern (Figure 3) segregated CeO2 particles are predominately observed. This seems to go in opposite direction and doesn’t correspond to the model illustrated in Figure 7b,f. On the other hand, a promoting effect in the Co3O4@CeO2-IE material compared to the Co3O4/CeO2-impreganted one is clearly observed from the catalytic data displayed in Figure 4, however I am wondering if this promoting effect could be simply attributed to the higher surface area of the former one. Both points need to be addressed, before considering this work suitable for publication.
Additional aspects to take into consideration:
1) According to the synthesis method of Co(OH)2, Mg(OH)2 is used. My concern is if Mg(OH)2 remains in the final material or if the authors did a post-treatment to remove this phase.
2) The colours used in the XPS figure (Figure 5a) need to be revised. In addition, the O1s core level is not as simple as the authors suggest. Contributions from Co(OH)2 (531.2eV) and hydroxyl groups (531 eV) beside the one corresponding to defects, need to be considered. In deed according to the authors 40% defects are observed in the Co3O4@CeO2-IE sample compared to 25% of the impregnated one. This is a huge difference not visualized from EPR analysis.
3) It is recommended to use the Scherrer equation in the XRD pattern to calculate particle size. Also, it would be interesting to calculate the unit cell parameters from the XRD pattern to support doping of Ce in the Co (OH)2 lattice.
4) Why activation energies have been calculated only in the low temperature range (25 to 111ºC) and not at all temperatures?. It is true that for that, the experiment has to be repeated ensuring differential conditions (i.e., Low conversion).
5) Some typos: Co2p3/2 instead of Co2p3/2; core line instead of emission when refering to “Co2p3/2” ; apparent activation energy instead of activation energy; what is the meaning of “These results suggest that Co3O4@CeO2-IE mainly exposes the CeO2 inserts into Co3O4 surface,?; what is the meaning of PM2.5 (VOCs from PM2.5)
6) According to the authors, the adsorption structure of propane on the two substrates are shown in figure 7a,b. But it has not been included. Also the Fermi level is not understood. In general, the DFT section is not clear and need to be discussed in more detail.
Round 2
Reviewer 1 Report
The manuscript was revised and responses sent are satisfied. Authors added and improved the text. Thus, I can recommend it for publication
Reviewer 2 Report
the authros have answered all the questions risen from the reviewers. The paper is suitable for publication